# Unraveling the processes shaping mammalian gut microbiomes over evolutionary time

Mathieu Groussin[1,2,*], Florent Mazel[3,*], Jon G. Sanders[4], Chris S. Smillie[1,2,5], Sébastien Lavergne[3], Wilfried Thuiller[3] & Eric J. Alm[1,2,5]

Whether mammal–microbiome interactions are persistent and specific over evolutionary time is controversial. Here we show that host phylogeny and major dietary shifts have affected the distribution of different gut bacterial lineages and did so on vastly different bacterial phylogenetic resolutions. Diet mostly influences the acquisition of ancient and large microbial lineages. Conversely, correlation with host phylogeny is mostly seen among more recently diverged bacterial lineages, consistent with processes operating at similar timescales to host evolution. Considering microbiomes at appropriate phylogenetic scales allows us to model their evolution along the mammalian tree and to infer ancient diets from the predicted microbiomes of mammalian ancestors. Phylogenetic analyses support co-speciation as having a significant role in the evolution of mammalian gut microbiome compositions. Highly co-speciating bacterial genera are also associated with immune diseases in humans, laying a path for future studies that probe these co-speciating bacteria for signs of co-evolution.

[1] Center for Microbiome Informatics and Therapeutics, Massachusetts Institute of Technology, Cambridge, Massachusetts 02139, USA. [2] Department of Biological Engineering, Massachusetts Institute of Technology, Cambridge, Massachusetts 02139, USA. [3] Laboratoire d Ecologie Alpine, CNRS, University of Grenoble Alpes, FR-38041, Grenoble Cedex 9, France. [4] Organismic and Evolutionary Biology, Harvard University, 26 Oxford St, Cambridge, Massachusetts 02138, USA. [5] The Broad Institute of MIT and Harvard, 415 Main Street Cambridge, Massachusetts 02142, USA. * These authors contributed equally to this work. Correspondence and requests for materials should be addressed to E.A. (email: ejalm@mit.edu).

The evolution of binary symbiotic relationships between animal hosts and individual microbial symbionts is well documented[1–3]. However, little is known about the mechanisms that shape community structure during the long-term symbiosis of hosts and their gut microbiomes. If sister host lineages share similar bacteria, gut communities may recapitulate host phylogeny. This pattern, referred to as 'phylosymbiosis'[4–6], does not imply a process and vertical inheritance of symbionts. We define 'vertical inheritance' as the restricted transmission of bacterial lineages within- rather than between-host lineages, as in dispersal exclusively to conspecifics; this could include but would not require transmission from mother to offspring. Phylosymbiosis may arise from the neutral vertical inheritance of symbionts (for example, co-speciation associated with allopatric speciations of hosts, see Supplementary Fig. 1), and/or from selective vertical inheritance, for instance, with intimate co-evolution between host and microbes[7–9]. But it can also arise independently of vertical inheritance, if closely related hosts with similar genetic or behavioral traits select similar bacteria from the environment[10] (Supplementary Fig. 1). The signal of phylosymbiosis can erode over time or even disappear, when a selective trait such as diet is decoupled from host phylogeny, promoting the horizontal acquisition of bacterial symbionts from the environment or from distantly related hosts[11,12].

In mammals, both at the intrahost and interhost species levels, whether host evolutionary history (host genetics and phylogeny, respectively) or host diet exerts a stronger influence in shaping the gut microbiome is controversial. Several studies[11,13–16] have claimed that diet is the main driver of gut microbiome composition. Some authors have claimed that host genetics has a minor contribution relative to environmental factors in humans and chimpanzees[15,17,18], while others have found evidence for a stronger impact of host genetics[19,20]. At the interhost level, it has recently been shown that two gut bacterial families possess lineages that harbour patterns of co-speciation in four hominid hosts[7], highlighting the influence of host phylogeny at short evolutionary scales. However, previous investigations have disagreed on whether or not[11,21] a signal for phylosymbiosis exists at larger evolutionary scales, across all mammalian gut microbiomes[4,11,21,22]. Because of all these uncertainties, and because we anticipate that correlation with host phylogeny would be generated by a mix of vertical and horizontal inheritance, while the correlation with diet would be primarily driven by horizontal inheritance, it is unknown whether mammalian gut microbiomes primarily evolve through vertical or horizontal inheritance over evolutionary time.

Here we characterized the variation in mammalian gut microbiome compositions in light of host diet and phylogeny. We are defining diet with a coarse granularity, using nine large dietary categories from the EltonTraits database[23]. A percentage for each species is assigned to each of the nine categories (see 'Host phylogeny and dietary data' in the Methods and Supplementary Table 1). Hence, this study focuses on the impact that large dietary shifts had on microbiomes at the scale of mammalian evolution. Furthermore, we employ 'host phylogeny' as a composite term that encompasses all traits that change roughly clock-like along the phylogeny of hosts and that might influence microbiome compositions, such as genetic or immunological factors[24]. We hypothesized that major changes in diet and host phylogeny may have driven vertical and horizontal inheritance of bacterial lineages at different bacterial phylogenetic scales[6,25]. For example, when mammalian lineages shifted their diet towards herbivory, they might have horizontally acquired herbivorous-specific bacterial lineages, and those bacterial lineages could even predate the divergence of the mammals. Furthermore, if host phylogeny shapes gut

microbiome compositions independently of the significant dietary shifts that occurred during mammalian evolution, and if vertical inheritance is generating this correlation with host phylogeny, these associations should be stronger in recent regions of the bacterial tree (as co-speciation events are not possible prior to the evolution of mammals). However, if vertical inheritance is not involved in generating this diet-independent correlation with host phylogeny, associations with host phylogeny should be seen at timescales of bacterial evolution that are decoupled from host evolution. For example, a given mammalian clade might select non-vertically inherited bacteria within a bacterial lineage that arose prior to the emergence of mammals.

Thus a phylogenetically informed approach incorporating compositional disparities along the bacterial phylogenetic timescale may allow us to disentangle the individual contributions of host phylogeny and major dietary shifts and to better understand how and to what extent the different types of bacterial inheritance (vertical and horizontal) have driven gut community evolution. Using such an approach, we show that diet drives the horizontal acquisition of bacterial lineages that belong to ancient bacterial clades and that host phylogeny predicts the presence of more recent bacterial lineages. We show that gut microbiomes have recorded the information of major dietary shifts that occurred during the evolution of mammals, allowing us to predict ancient diets from the reconstruction of ancient microbiomes. Associations between microbiome compositions and host phylogeny are universal in mammals and stronger among recently diverged mammals. Finally, our results suggest that co-speciation between bacterial lineages and their mammalian hosts partly drives these patterns of phylosymbiosis.

## Results

**Phylogenetic decomposition of community dissimilarities.** Gut microbiome content in operational taxonomic units (OTUs) can vary greatly between two communities (that is, hosts). Usually, these compositional dissimilarities (β-diversity[26]) are described with a single measure, either using taxonomic metrics, such as the Sørensen metric[27], or using phylogenetic metrics, such as UniFrac[28]. However, relying on a single number to describe community dissimilarities may be too simplistic if different factors shape compositions in OTUs at different bacterial phylogenetic scales[25]. Yet, there is no robust framework in the literature that integrates compositional β-diversity along a phylogenetic timescale.

We developed a new method (BDTT; for β-diversity through time) to account for this potential temporal scale disparity between factors, here to separate the influence of host diet and phylogeny on gut microbiota compositions along the timescale of bacterial evolution (Supplementary Fig. 2). BDTT computes compositional turnover (using Sørensen or Bray–Curtis metrics, see Methods) between communities at different time period along the bacterial phylogenetic timescale, producing a profile of β-diversities[29]. From the leaves to the root, the bacterial tree (here, reconstructed de novo from all 16S rRNA reads) is continuously sliced, either by time or evolutionary distance. For each time period, the tree of bacteria is cut, yielding clades that can serve as OTUs in downstream analysis. Microbiome composition is then determined for each mammalian species in terms of these new OTUs, and pairwise compositional dissimilarities are computed. Importantly, the BDTT profile provides a phylogenetic decomposition of the broadly used UniFrac metric[28] (Supplementary Fig. 2 and Supplementary Note 1), Finally, at each time period, β-diversities are correlated to host diet and phylogenetic distances, and the comparison of the amount of β-

diversity explained by each factor across bacterial timescales may reveal the phylogenetic levels at which their individual influence is greatest. We run BDTT on simulated data sets (see 'Validation of BDTT on simulated data' in the Methods) and we show that it is able to disentangle the effect of different factors shaping community assembly at different phylogenetic scales (Supplementary Fig. 3 and Supplementary Note 2).

Note that, for BDTT to be informative, it is not necessary that bacterial time estimates (in millions of years ago (Myr ago)) be strictly accurate, but rather that the relative order of branching is conserved from relatively ancient to relatively modern divergences. Branch lengths expressed in the expected number of substitution per site were also used to approximate time and to cluster sequences into OTUs at different slices in our BDTT approach, yielding to identical patterns (Supplementary Figs 2 and 4).

We applied BDTT to a data set of 33 mammalian gut microbiomes[11], composed of 44,444 dereplicated and chimera-free amplicons of the 16S rRNA gene (V2 region), from which we reconstructed the time-calibrated bacterial phylogenetic tree. Host phylogenetic distances between our 33 mammals were deduced from a time-calibrated ultrametric phylogenetic tree of mammals[30], which was recently updated[31,32]. Diet distances were deduced from EltonTraits[23], a database that compiles dietary attributes for all mammalian species.

**Phylogeny and diet shape microbiomes at different scales**. Consistent with our hypothesis, BDTT was able to disentangle the effects that the major dietary shifts had on gut microbiome compositions from those of host phylogeny. This was possible because host phylogeny shapes gut microbiome compositions mainly near the leaves of the bacterial tree (Mantel test; $R^2 = 0.38$ at 100 Myr ago, $P$ value < 0.001; $R^2 = 0.03$ at 1,000 Myr ago, $P$ value > 0.05), while host diet mostly determines the distribution of more ancient bacterial lineages among hosts (Mantel test; $R^2 = 0.08$ at 100 Myr ago, $P$ value < 0.001; $R^2 = 0.22$ at 1,000 Myr ago, $P$ value < 0.001) (Fig. 1a,b).

We performed multiple control experiments to evaluate the robustness of this signal of phylogenetic scale disparity between host phylogeny and diet (Supplementary Figs 4–7 and Supplementary Note 3 and 4). We measured the influence of the intrahost species variability of microbiome compositions, the influence of topological uncertainties in the tree of bacteria and the influence of differences in statistical power at different depth of the bacterial tree. We also used branch lengths in the bacterial tree expressed in average substitution/site as a proxy of time in the BDTT approach. We controlled for the influence of the unequal sampling size across samples by performing rarefaction of the OTU tables, and we controlled for the differences in granularity between the host phylogenetic and dietary distance matrices. All of these experimental controls support that our analysis uncovers a genuine and robust biological signal of scale disparity between the two factors. Finally, we described the process of mammalian diet evolution and then used this process to simulate traits along the phylogeny of hosts. We used these simulated traits to run BDTT and compared the simulated correlation profiles with the correlation profile obtained when using observed diets. We show that the peak of correlation between microbiome compositions and diet at ancient timescales is not solely an echo of host phylogeny and conclude that the contribution of diet on microbiome compositions is partially decoupled from the host phylogenetic history (see Supplementary Fig. 8 and 'The contribution of diet on microbiome compositions is partially decoupled from the host phylogenetic history' in Supplementary Note 3).

As mammals diverged recently in comparison to bacteria, the effect of host phylogeny on recent bacterial lineages is consistent with, although not direct proof of, co-diversification between hosts and their gut microbiomes. The effect of diet on more ancient bacterial lineages is consistent with the evolutionary conservation of functional traits involved in the digestion of dietary compounds in ancient bacterial taxa[25,33]. Note that BDTT allows us to statistically disentangle the temporally separated portions of the contributions of host phylogeny and diet (when defined with a coarse granularity), not the totality of the processes themselves.

To further resolve the roles of host diet and phylogeny, we asked whether the more recent bacterial lineages that correlate with host phylogeny are nested within ancient diet-related bacterial lineages or whether they are independent from them (Supplementary Fig. 9a,b). After removing the bacterial lineages correlated with broad categories of diet (Fig. 1a), we found that the correlations between host phylogeny and bacterial composition within recent time slices decreased but remained very strong (Fig. 1c), close to their previous level (Mantel test; $R^2 = 0.34$ versus 0.38 at 100 Myr ago). This means that although some bacteria related to host phylogeny at recent timescales are nested within higher clades also related to broad categories of host diet, a large part of the bacteria associated with host phylogeny are different from the bacteria associated with diet (see also 'Bacterial lineages correlating with host phylogeny and diet lowly overlap' in Supplementary Note 5). We also measured to what extent the covariation between host phylogeny and diet drives gut community β-diversities. We observed that, at all bacterial phylogenetic scales, the covariation between host phylogeny and diet only weakly explains bacterial community dissimilarity ($R^2$ ranging from about 0.01 to about 0.04, see Supplementary Fig. 9c and 'The covariance between host phylogenetic and dietary distances poorly explains community dissimilarities' in Supplementary Note 5). Altogether, these results suggest that, even though broad shifts in diet can be locally correlated with host phylogeny[34,35], the impacts of these factors on changes in the microbiome can be primarily observed in different microbial lineages.

**Herbivory and carnivory act at distinct timescales**. We also discovered a timescale disparity between herbivore- and carnivore-associated bacterial lineages. Herbivory is associated with bacterial lineages that arise deeper in the bacterial phylogeny (> 200 Myr ago) than those associated with carnivory, which are confined in a limited range of phylogenetic scales (between 150 and 300 Myr ago, see Fig. 1d). This suggests that herbivory and carnivory are associated with bacterial lineages that emerged at different evolutionary timescales and that traits allowing bacterial lineages to thrive within a mammalian herbivorous gut appeared early in bacterial evolution, while those permitting specialization in carnivore guts arrived much later, which is consistent with the late appearance of carnivorous animals[36].

**Omnivores do not harbour omnivore-specific bacteria**. The fact that some bacterial lineages may have functionally adapted to thrive in gut environmental conditions that are diet-specific and not host-specific raises the question: do generalist hosts harbour distinct microbes that are also 'generalists' and can digest both plant and animal material or do they include a mixture of herbivorous and carnivorous bacteria? Upon closer inspection, we found specialist bacterial lineages associated with both herbivores and carnivores (Fig. 2a), suggesting functional adaptations to these two types of diet-related gut environments. However, we did not find specialist bacteria associated with omnivores,

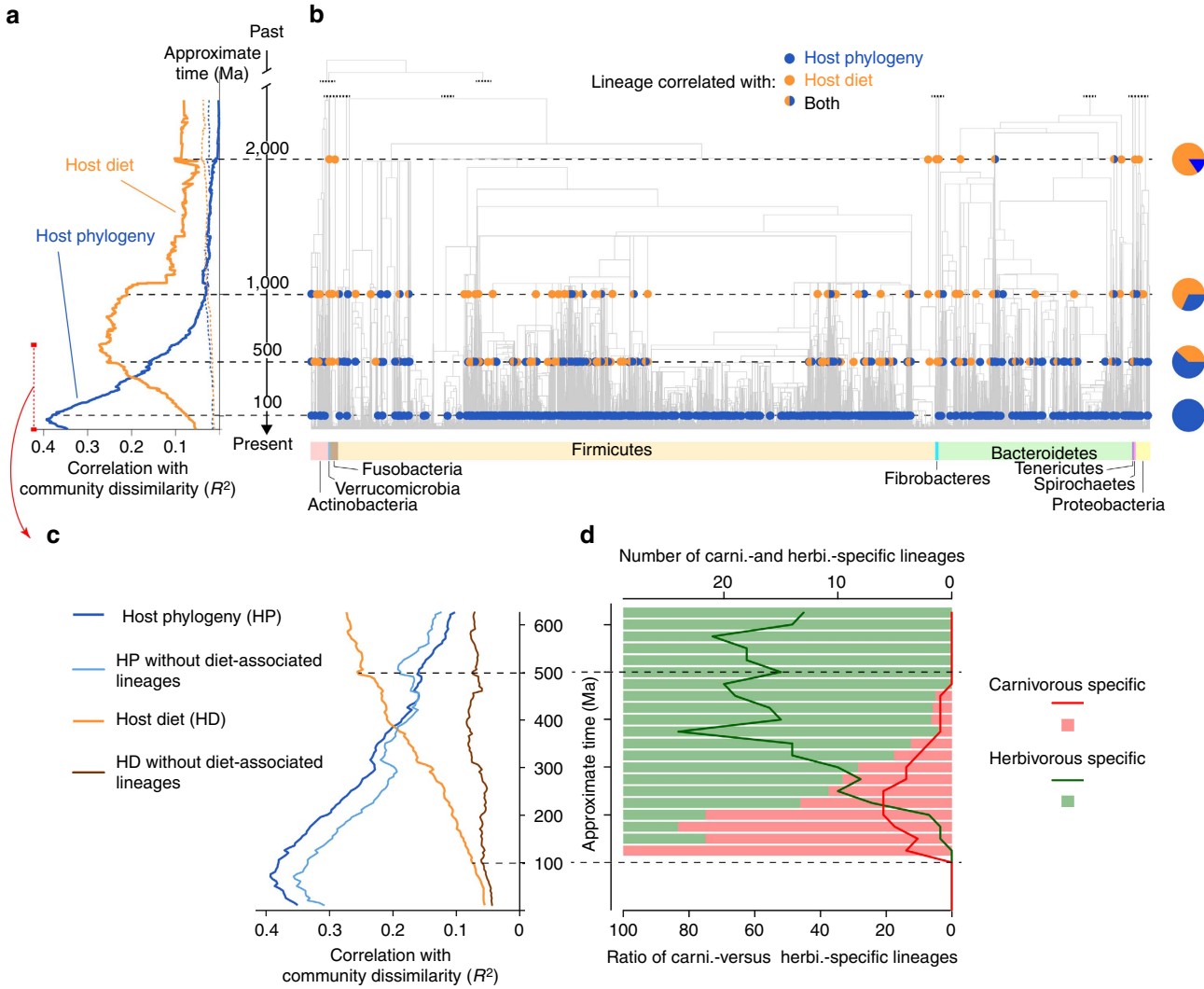

**Figure 1 | Phylogenetic-scale disparities in mammalian gut microbiomes.** (**a,b**) Bacterial lineages that diverge recently in evolutionary history show high levels of correlation with host phylogenetic distances (blue) while correlation with host dietary distances (orange) is greatest for more ancient lineages. (**a**) Lines show correlations between the pairwise Sørensen compositional dissimilarities and pairwise host phylogenetic or dietary distances (dashed lines: 95% null envelope). (**b**) Individual bacterial lineages correlated with diet or phylogeny (circles). Pie charts represent the percentage of lineages that correlate significantly with each factor at different times. The phylogenetic scale is common to plots **a,b**. (**c**) When diet-associated lineages (in **b**) are removed, correlation with host phylogeny (dark blue) still holds (light blue), but correlation with host diet does not (orange versus brown). (**d**) Each bacterial lineage having a significant correlation with diet (**b**) was called herbivorous- or carnivorous-specific if it is only found in herbivores or carnivores, respectively. Herbivory is associated with bacterial lineages that arise earlier in bacterial evolution than those associated with carnivory.

suggesting that omnivores contain a combination of herbivorous and carnivorous bacterial groups. Figure 2a shows these results using a joint projection of the diet-associated bacterial lineages with hosts that are separated by diet on the same ordination space (see also Supplementary Note 6).

**Gut microbiomes can predict ancient mammalian diets.** We next investigated whether the presence of these diet-specific bacteria is sufficient to predict diet in extant hosts. To test this, we built a multinomial regression model (see Methods and Fig. 2b) to predict diet in mammalian hosts from their microbiome composition. We evaluated the accuracy of our model using cross-validation experiments, and we show that our microbiome-based method is accurate and performs well at estimating host diet (Supplementary Fig. 10a–c and 'Accuracy of our microbiome-based method of diet prediction' in Supplementary Note 7).

As we could predict host diet, we hypothesized that we might be able to infer the diet of ancestral mammals by reconstructing their microbiome. We further hypothesized this should only be possible if we use an appropriate phylogenetic cutoff (as before, ~300 Myr ago or ~94% OTUs) to describe community structure. In particular, reconstruction of widely used 97% OTUs should not be able to reconstruct ancestral dietary patterns. To test our hypotheses, we reconstructed ancestral microbiome compositions for each mammalian ancestor using a maximum likelihood (ML)-based approach[37], that accounts for the vertical and horizontal inheritance of OTUs along the host phylogeny. We used a model that integrates over the numerous events of gain and loss of bacterial lineages that occurred during the millions of years of host evolution, with varying rates across host lineages. We then used our microbiome-based model to predict ancestral diets from these ancestral microbiomes. We compared our microbiome-based predictions to those obtained with a classic macro-evolutionary, trait-based model that considers diet as a

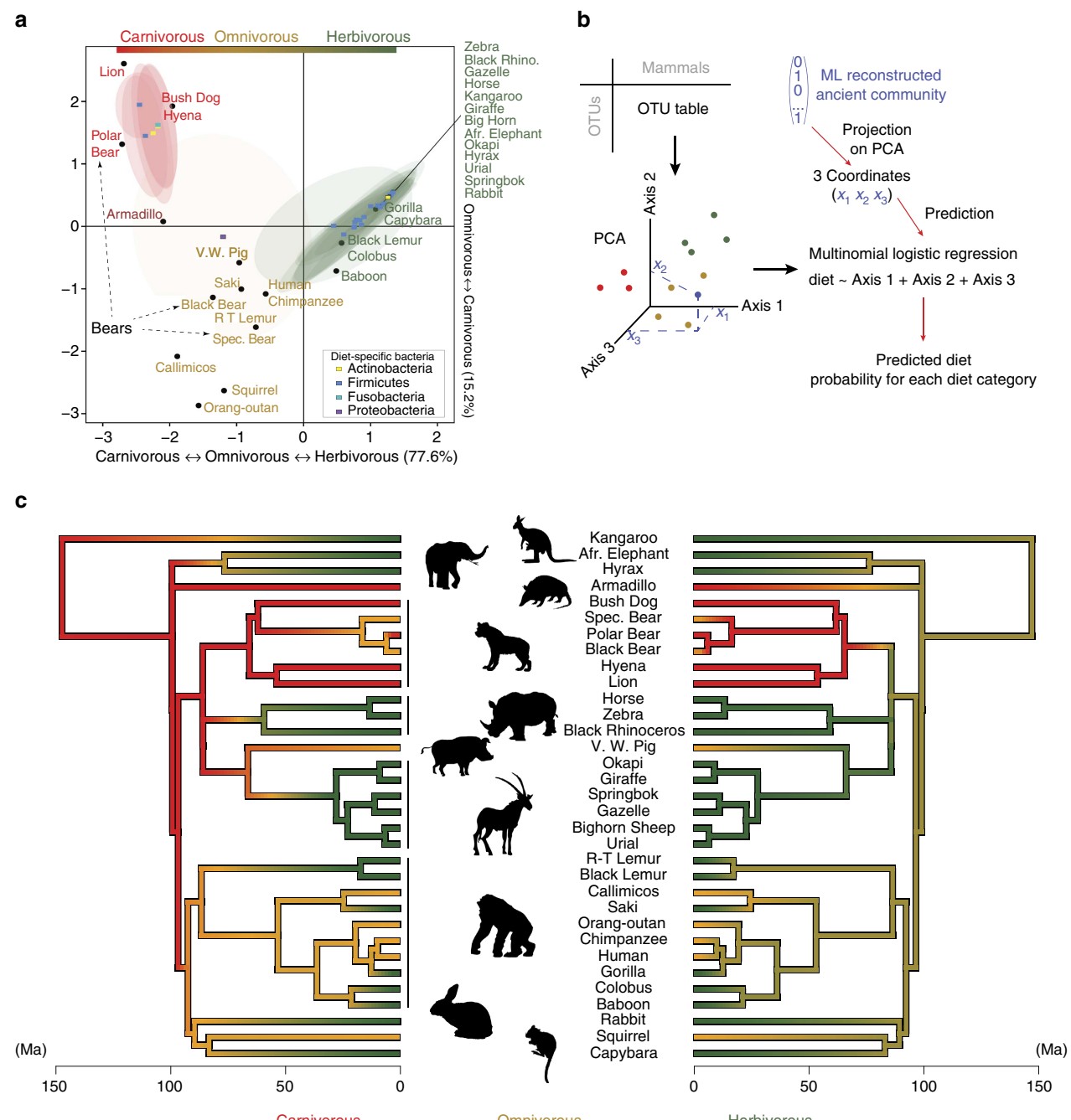

**Figure 2 | Gut microbiomes predict extant and ancestral mammalian diets.** (**a**) Diet-correlated bacterial lineages (squares coloured by phylum) and mammals (black dots) separated by dietary distances are jointly projected in the same ordination space using the OMI[59] mapping procedure (see Methods). Predicted niche breadth[59] of a bacterial lineage is depicted with an ellipse. The time period selected to define bacterial lineages is ~300 Myr ago. Bacterial niche breadths show that many bacterial groups are herbivore or carnivore specific. However, no bacterial lineage has a niche breadth encompassing omnivores only (omnivores are in brown). (**b**) Workflow for microbiome-based prediction of ancient diet. (**c**) Inference of ancient mammalian diets. Left: trait-based reconstruction using 1,534 mammals[34]. Right: Microbiota-based reconstruction with the 33 mammals under study. At each ancestor, the probability vector for each diet category is transformed into a linear variable bound between 0 (carnivorous) and 1 (herbivorous). Between two ancestors, diet is assumed to evolve linearly. Animal images courtesy of Julien Renaud.

discrete variable with three different states (Carnivorous, Omnivorous and Herbivorous). We performed a trait-based diet reconstruction with (i) the present mammalian data set (33 mammals) and (ii) a much larger taxonomic sampling (1,534 species of mammals[34]). This large set of species more exhaustively samples mammalian diversity and previously provided trait-based diet reconstructions that are in agreement

with the fossil record[38]. For these reasons, we considered this latter reconstruction as a reference, to which microbiota-based reconstructions were compared. Interestingly, we found that microbiome-based inferences agreed at 70% of ancestral nodes to these reference trait-based predictions (Fig. 2c). We observed that transitions towards herbivory are associated with multiple and convergent horizontal gains of herbivorous-specific bacterial

clades across several branches of the mammalian phylogeny (Supplementary Fig. 11 and Supplementary Note 8). Most of the divergent predictions, as shown in Fig. 2c, are located in ancient, poorly sampled areas of the mammalian tree (carnivorous Chiroptera, Afrotheria, Xenarthra and Marsupialia are absent in the present data set, see 'Microbiome-based and trait-based inference of ancestral diets in mammals' in Supplementary Note 7).

We further compared the precision of the predictions obtained with the microbiome- and trait-based methods on a similar taxonomic sampling (33 species). We found that the entropy of the ancestral dietary probability distributions, which measures the uncertainty in these distributions, is significantly lower with microbiome data (Wilcoxon test, $W = 232$, $P$ value $= 0.0002$), indicating a higher precision of the microbiome-based method (Supplementary Fig. 10d–f and 'Microbiome-based and trait-based inference of ancestral diets in mammals' in Supplementary Note 7). Finally, bacterial phylogenetic groups summarized at 120 Myr ago (or $\sim$97% 16S similarity) or 600 Myr ago ($\sim$91% similarity) were much less accurate at predicting diet (see 'Reconstruction of ancient diets from microbiomes defined at alternative bacterial phylogenetic scales' in Supplementary Note 7). These results underscore the importance of identifying the correct phylogenetic resolution with which to define communities, in order to study patterns of host–microbiome evolution. Overall, our results show that gut microbiomes record the information of past dietary adaptations with the horizontal acquisition of diet-specific bacteria and that signals of these adaptations still remain in the microbiomes of extant mammals.

**The phylosymbiosis signal is strong in mammals.** Phylosymbiosis is a pattern describing higher compositional similarity (that is, low β-diversity) between bacterial communities colonizing closely related hosts compared with distantly related hosts[4–6]. The correlation between microbiome composition and host phylogeny at shallow bacterial phylogenetic scales observed in Fig. 1 does not universally describe the variation and magnitude of phylosymbiosis across the host mammalian tree:

the signal for phylosymbiosis might be uniform or variable across mammalian clades, and it might be weak or strong. In the following, we address these questions with communities defined at a recent bacterial phylogenetic scale ($\sim$120 Myr ago, OTUs with $\sim$97% similarity, see Methods), where correlation with host phylogeny is high.

After $>$100 millions of years of mammalian evolution, many gains and losses of bacterial lineages occurred, possibly randomizing the genuine signal of compositional change that mirrors the mammalian phylogeny. To measure phylosymbiosis, we modelled the dynamics of gain and loss of bacterial lineages using the same phylogenetic model[37] that we previously used in the section 'Gut microbiomes can predict ancient mammalian diets'. We quantified phylosymbiosis with a hierarchical approach along the tree of mammals, by reconstructing ancestral communities for each mammalian ancestor (that is, for each node of the host phylogeny). We reasoned that if the size of a reconstructed community for an ancestor of a given mammalian clade is higher than randomly expected under a null model in which host–bacteria associations are shuffled, then mammals belonging to this clade harbour a phylosymbiotic signal, because their gut microbiota share more bacterial lineages between each other than randomly expected. We computed the magnitude of the phylosymbiosis signal along the mammalian host tree, for each ancestor, by computing a standard effect size (SES) per branch/node:

$$\text{SES}_{\text{phylosymbiosis}} =$$
$$\frac{\text{Size ancestral OTU content}_{\text{Obs}} - \text{mean(size ancestral OTU content}_{\text{Null}})}{\text{s.d.(size ancestral OTU content}_{\text{Null}})}$$

$$(1)$$

For a given ancestor, the higher the SES is, the stronger the signal for phylosymbiosis is. Although previous parsimony and distance-based methods provided conflicting results on the existence of a phylosymbiosis signal[4,11], we observe that most of the mammalian clades, both young and ancient (up to 80 Myr ago), harbour a significant (permutation test, $P$ value $<$0.05) and strong phylosymbiosis signal (Fig. 3a). The magnitude of

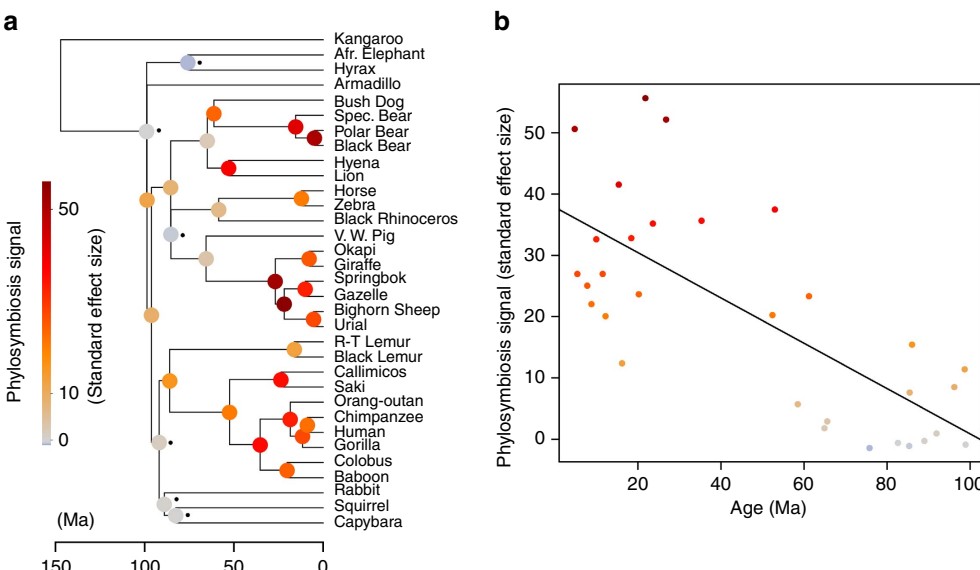

**Figure 3 | Strong phylosymbiosis signal in mammals. (a)** Long-term persistence of host–microbiota associations in mammalian evolution. Mammalian phylogeny with internal nodes coloured according to the degree of possession of clade-specific compositions of bacteria (clade-specific phylosymbiosis signal) (see Methods). Black dots denote mammalian clades that do not harbour a significant phylosymbiotic signal (Permutation test, 999 permutations, $P$ value $>$0.05). The weak phylosymbiosis signal in Rodents or Afrotheria is likely due to a poor sampling within these two groups. **(b)** Decay of the phylosymbiosis signal with time ($x$ axis: age of the corresponding nodes).

phylosymbiosis (SES, equation 1) decays with evolutionary time (Fig. 3b), consistent with multiple turnovers over millions of years of host evolution and/or with the long-term evolution of host traits, progressively selecting for different bacteria. Contrary to some previous conclusions[11], this turnover of gut microbes during mammalian evolution is not too rapid to erase the phylosymbiosis signal.

At the shallow phylogenetic scale at which we measured phylosymbiosis, the correlation between community dissimilarities and host dietary distances is weak but significant (Fig. 1a). We measured to what extent the phylosymbiosis signal is affected by dietary shifts. At each focal node in the mammalian phylogeny, we estimated the magnitude of dietary shift after the divergence of the two descendant lineages. We used our trait-based ancestral diet reconstructions using 1,534 mammalian species and measured the average diet distance with the two descendant nodes. As phylosymbiosis is negatively and linearly correlated to node ages (Fig. 3), we compared for each focal mammalian node the residuals of this regression to the magnitude of dietary shift at this node. If ancestral dietary shifts have an impact on phylosymbiosis, we expect to observe a negative correlation between phylosymbiosis residuals and the magnitudes of dietary shift. We show that the effect of dietary shift poorly explains the decrease in phylosymbiosis compared with what would be expected by age: the regression has a negative but weak slope ($R = -0.15$, Supplementary Fig. 12). We conclude that dietary transitions along mammalian evolution had minor effect on the part of the microbiota that correlates to host phylogeny.

**Co-speciation between hosts and their gut bacterial symbionts**. Phylosymbiosis does not necessarily imply vertical inheritance, it only reflects congruency between gut bacterial compositions and host phylogeny. We next asked whether this signal could have been generated through vertical inheritance, via co-speciations between symbionts and mammals. Besides co-speciation, at least two other possible mechanisms can generate the same signals of correlation with host phylogeny and of phylosymbiosis[10] (Fig. 3a and Supplementary Fig. 1): horizontal inheritance through host swaps within related hosts, and environmental filtering by closely related hosts that select similar symbionts from the environment. Of these various mechanisms, co-speciation is more likely to result in congruent splits between the symbiont and host phylogenetic trees. We used a probabilistic model of host tree/symbiont tree reconciliation implemented in ALE[39–41] to measure the magnitude of topological congruence between symbiont and host trees and the number of bacterial lineages experiencing more co-speciation than host-swap events.

Interestingly, we find that a majority of bacterial clades (67%) harbour more co-speciation than host-swap events (Fig. 4a,b). To test whether this percentage is higher than one would expect by chance only, we evaluated the percentage of bacterial clades that would show more co-speciation that host swap under a null model in which phylogenetic relationships between hosts are disrupted. We observed that, by chance, we would obtain about 10% of OTUs with more co-speciation than host-swap events, much less than the estimates obtained with the observed biological data.

Even though OTUs may have more co-speciation than host-swap events, the co-speciation rate might not be significantly higher than the host-swap rate. This might be due to insufficient phylogenetic information in the 16S rRNA data. In addition, an OTU might have more co-speciation than host swap only because the reconciliation algorithm (ALE) overfitted the host tree when searching for the best scenario of bacterial evolution, leading to

an overestimation of the number of co-speciation events. To control for these effects for each individual OTU, we evaluated whether the observed rate difference between co-speciation and host swap is significantly higher than expected under a null model (see Methods). We found that, within the 67% of bacterial clades showing more co-speciation than host swap, only 16% show no significant rate differences (Fig. 4a).

Among the OTUs that have a distribution across hosts that correlates with host phylogeny, 89% have more co-speciation than host swap (Fig. 4a,c), supporting that co-speciation is likely to be a major driver of the correlation with host phylogeny. Moreover, 31% of these bacteria show no sign of host swap and have fully congruent phylogenies with the host phylogeny. It is possible, however, that iterative bacterial specializations on related host linages (or host-swap speciation) locally yield concordant nodes in symbionts/host topologies, inflating the amount of detected co-speciations[42] (see Supplementary Discussion).

We did not observe that strong changes in diet are driving the co-speciation signals that we measured. Indeed, when running reconciliations considering mammalian herbivores only, we observed that the frequency of all OTUs having more co-speciation than host swap was similar to the frequency observed with all mammals (Z-test, 70% versus 67%, $P$ value $> 0.05$). However, it does not rule out the possibility that fine-grained differences in diet that are locally correlated with host phylogeny and, which are not captured by our diet distance metric, have promoted co-speciation or, alternatively, have created spurious co-speciation signals (see Supplementary Discussion).

Altogether, while most bacterial lineages do show evidence of horizontal inheritance, our results also suggest that vertical inheritance is an evolutionary path followed by numerous mammalian gut symbionts and extend the recent results on patterns of co-speciation between gut bacterial lineages and Hominids[7] to larger evolutionary scales (see Supplementary Note 9).

**Co-speciating bacteria are associated with disease in humans**. We next investigated the characteristics of bacterial lineages that harbour high co-speciation levels. Interestingly, at the scale of mammals, we found that many widely studied genera showed low levels of vertical inheritance with hosts (Fig. 4d, Supplementary Fig. 13 and Supplementary Note 9). These findings are consistent with the notion that these metabolically important and cosmopolitan organisms provide similar and probably diet-related functions across diverse hosts and thus are more likely to be transferred across hosts or acquired from the environment with little host selectivity. By contrast, we reasoned that organisms specifically and tightly linked to host physiology (for example, via immunity) might be more likely to be restricted to their host. In support of this hypothesis, we find that 13 out of the 20 bacterial lineages that inhabit humans and have high co-speciation rates ($> 0.8$) belong to five different genera (such as *Subdoligranulum*) that are strongly associated with a human immune disease, inflammatory bowel disease (IBD)[43] (Fig. 4e and Supplementary Fig. 14). This enrichment in highly co-speciating OTUs among genera negatively associated with IBD in comparison with lowly co-speciating OTUs is strongly significant (permutation test, $P$ value $= 0.0015$, see Supplementary Note 9). All five of these genera are strongly depleted in patients affected by both Crohn's disease and ulcerative colitis[43], perhaps suggesting a functional link between co-speciating bacteria and host immune function. It should be noted, however, that even among published IBD studies there can be differences between disease-associated taxa,

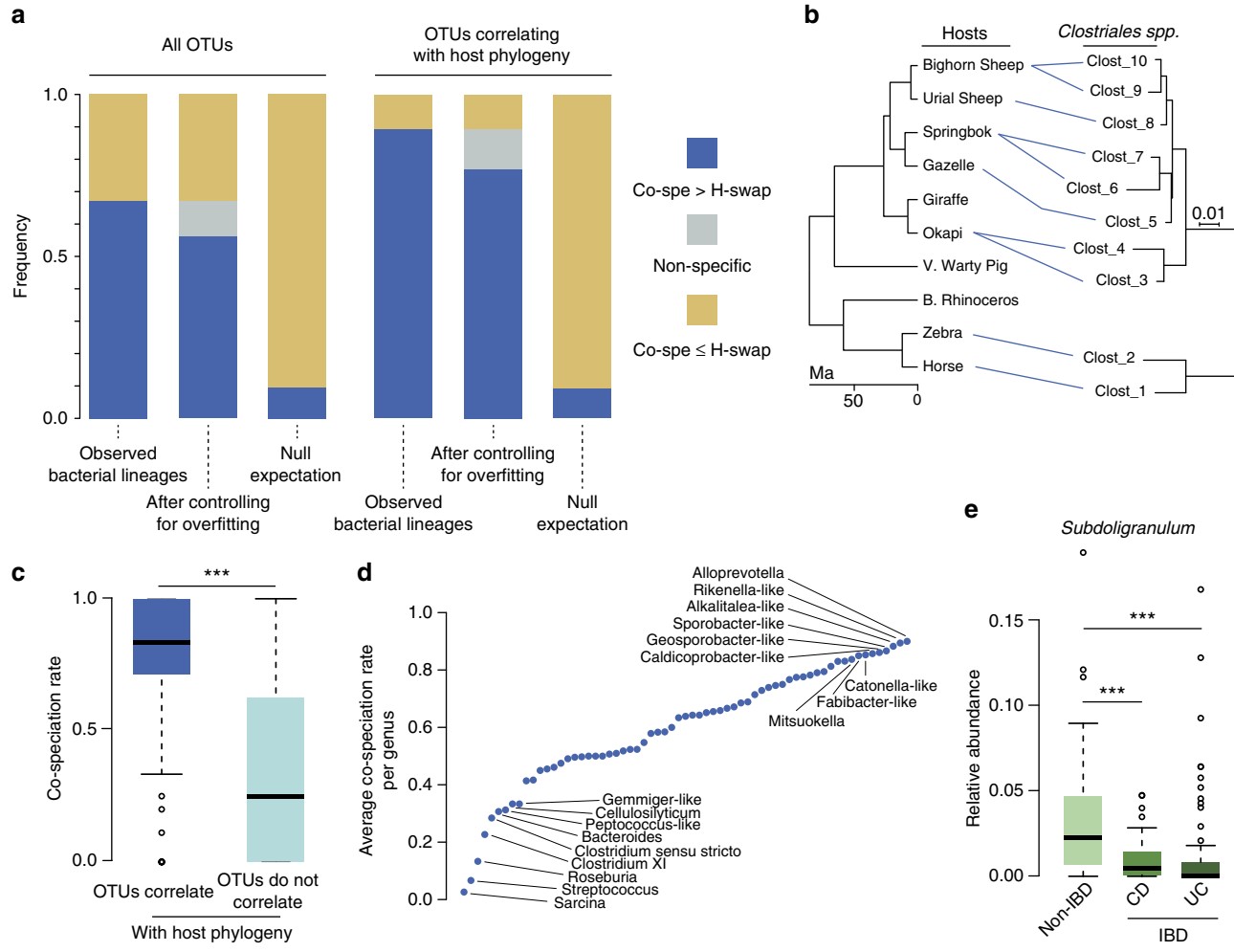

**Figure 4 | Large-scale vertical inheritance of mammalian gut symbionts.** (**a**) Frequencies of bacterial lineages harbouring either more co-speciation (Co-spe.) or more host-swap (H-swap) events are shown, either for all OTUs or only for OTUs that are correlated with host phylogeny. Out of the 350 bacterial lineages correlated with host phylogeny and present in at least four hosts, 313 harbour more co-speciation events than host-swaps. Non-specific (grey) lineages are lineages with a non-significantly higher rate of co-speciation than the rate of host-swap. This higher observed rate may be due to overfitting to the host tree (see Methods). The Null Expectation bar represents the expected frequency of lineages harbouring more co-speciation than host-swaps by chance (see Methods). (**b**) Example of a co-speciating bacterial lineage belonging to the Clostridiales order. A blue line indicates the presence of a symbiont in a host. (**c**) OTUs correlated to host phylogeny harbour higher co-speciation rates than OTUs that are not. Co-speciation rate per OTU is defined as the amount of co-speciation events relative to the number of host-swap events (two-tailed Wilcoxon's rank-sum test, ***P value < 0.001). (**d**) Average co-speciation rate per bacterial genus. For a full list of genera, see Supplementary Fig. 13. (**e**) *Subdoligranulum* has high co-speciation rates and is correlated with IBD in humans (two-tailed Wilcoxon's rank-sum test). CD, Crohn's disease; UC, ulcerative colitis. Other genera with similar patterns are shown in Supplementary Fig. 14.

so future studies are needed to confirm these preliminary observations and to turn these intriguing correlations into proper demonstration of causation.

## Discussion

Our findings are relevant to the recent debate over the extent to which hosts and their associated microbial communities should be considered as holobionts, defined as coherent genomic systems on which selection operates[44,45]. However, as was recently pointed out[10], evidence for a tight association between a small number of taxa and their hosts does not necessarily generalize to the entire microbiome, and we find that only a minority of lineages fully co-speciate with their hosts. Moreover, evidence for co-speciation does not necessarily imply co-evolution, which needs to be established through more in-depth and mechanistic studies. Nonetheless, our findings reveal the specific microbial

lineages most promising for such detailed study which are not the bacterial groups that have been the focus of most prior work and suggest their relevance to human health and disease.

## Methods

**Mammalian microbiome data set.** We analysed a data set of mammalian gut microbiota containing 33 mammals[11] that represent 10 mammalian orders and cover various diets (carnivorous, herbivorous and omnivorous)[30]. These data are ideal because they cover a large diversity of mammalian clades and were produced at the same time within a single study, avoiding biases introduced with variation in DNA extraction protocols, choice of 16S primers and sequencing platform. The data consist of amplicons of the V2 region of the 16S rRNA gene. As conspecific samples were originally available in this data set for seven host species, only one individual per species was retained for further analyses (and chosen at random) to focus on the evolution of gut microbiota at the interspecies level only. We also ran analyses including the other conspecific samples to control for the effect of intrahost species variability in gut microbiome compositions. The number of hosts in this data set (33 mammals) did not preclude us from obtaining strong statistical

support and power to discriminate between alternative hypotheses in all our analyses.

We used USEARCH v8.0.1517 (ref. 46) to dereplicate 16S sequences. Chimera sequences were removed from the data, using the UCHIME algorithm[47] and the reference Gold database, containing 5,181 16S sequences. We subsequently used the *de novo* algorithm, also implemented in UCHIME, to further remove chimeras. We used the Ribosomal Database Project Classifier[48] to assign a taxonomy to each unique 16S sequence. We retained bacterial sequences with an assignment probability $> 0.8$ at the phylum level (85% of all sequences). The final data set comprises 44,444 unique bacterial sequences.

**Phylogenetic reconstruction.** To use BDTT on our data, we reconstructed the phylogenetic tree of all 16S unique sequences. First, we added the V2 region of the 16S rDNA genes from 139 Cyanobacteria, 115 Rickettsiales, 21 Chlorobium and 24 Chromatiales to the set of 44,444 stool sequences. These external 16S sequences were used to incorporate calibrations when computing the divergence times (see below) and were subsequently removed for all our analyses.

We aligned all 16S reads with the sina program (version 1.2.11) using the Silva database[49], producing an alignment of 751 sites. We then removed sites containing $> 95\%$ of gaps from the alignment. The final alignment contains 257 positions. To limit computational burden, we used FastTree[50] to reconstruct the ML phylogenetic tree of all 44,444 unique stool sequences plus 299 external sequences. We used the GTR model and the default CAT approximation to model rate heterogeneity across sites. We used the RDP taxonomy to constrain the topology of the phylogeny, forcing all phyla and all classes within the Proteobacteria (alpha, beta, gamma, delta, epsilon) to be monophyletic. The phylogenetic tree was rooted either on the branch separating Actinobacteria or Firmicutes from the rest of the sequences. Finally, we used PATHd8 (ref. 51) to produce a cladogram from the ML tree reconstructed by FastTree (see 'Phylogenetic reconstructions' in the Supplementary Methods for further details).

**β-diversity through time and phylogenetic clustering.** As explained in the section 'Phylogenetic decomposition of community dissimilarities' in the Results, we computed compositional turnover (using Sørensen[27] or Bray–Curtis[52] metrics) between communities at different time periods along the bacterial phylogenetic timescale. However, BDTT can be used with any type of taxonomic diversity metrics, such as Jaccard or Jensen-Shannon. BDTT can be employed on communities of both microorganisms and macroorganisms, in order to study the phylogenetic/timescale-dependent effect of a given factor on the distribution of biological entities across host sites or spaces.

**Validation of BDTT on simulated data.** To test whether the BDTT approach is able to disentangle the effect of factors shaping community assembly at different phylogenetic scales, we carried out simulation experiments. The rationale is to assemble communities under known processes, here under two independent environmental filters. Species are filtered out of communities according to their two 'environmental preferences' or 'traits' (each trait represents a particular environmental variable). We make these traits evolve with decreasing evolutionary rates over time under models that differ in the strength of the rate decrease (see below). More specifically, one trait displays phylogenetic signal towards the tip of the phylogeny while the second trait will carry phylogenetic signal in higher regions of the tree. If communities are filtered according to these two traits at the same time, we expect BDTT to be able to discriminate the phylogenetic scale at which each environmental gradient primarily shapes community composition (for example, the effect of the environmental variable that harbours deep phylogenetic signal should be seen at high phylogenetic scales in the BDTT analysis).

The simulation experiment contains four steps: (i) we simulated a phylogeny of 200 species. (ii) We simulated traits (representing the environmental preference of species) along this phylogeny with different models that have scale-dependent rates of trait evolution, that is, producing traits with phylogenetic signals that are located at different phylogenetic scales. (iii) We assembled communities of species with two environmental gradients that filter species according to their traits ( = environmental preferences). These two gradients are linked to two independent environmental preferences of species ( = two traits) that show phylogenetic signal at different phylogenetic scales (as simulated in (ii)). (iv) We applied BDTT to these data sets and tested whether a phylogenetic-scale disparity was observed (for further details, see 'Validation of BDTT on simulated data' in the Supplementary Methods).

**Host phylogeny and dietary data.** The 33 hosts in the data set[11] represent 10 mammalian orders and cover various diets (carnivorous, herbivorous and omnivorous)[30]. Host phylogenetic time distances between our 33 mammals were deduced from a time-calibrated ultrametric phylogenetic tree reconstructed with 4,510 species[30] and that was subsequently updated to include 5,020 species[31]. Without adding new mammalian species, we further updated the phylogenetic relationships and divergence times within the Carnivora clade with a highly resolved supertree that was recently published[32]. We used the EltonTraits 1.0 database to compute dietary distances[23]. EltonTraits compiles dietary attributes for a large number of mammals, including the 33 species present in our data set. Nine

dietary items (Invertebrate, Vertebrate (excluding fishes), Rotting carcass, Fish, Unknown Vertebrate, Fruits, Nectar, Seed, other plant materials (for example, grass, ground vegetation, seedlings, weeds, lichen, moss, small plants, reeds, cultivated crops, forbs, vegetables, fungi, roots, tubers, legumes, bulbs, leaves, above ground vegetation, twigs, bark, shrubs, herbs, shoots, aquatic vegetation, aquatic plants)) are defined and each species is assigned a percentage for each item depending on its diet. We used Euclidian distance to build the diet distance matrix. As our dietary categories represent fuzzy variables (proportions between 0 and 1), we also computed diet distances using the Manly index, implemented in the dist.ktab function in the 'ade4' R package[53]. This control revealed no impact on our results, so we only present results obtained with the Euclidian distance matrix.

**Measuring correlation profiles with diet and host phylogeny.** For each time period, we correlated β-diversities to host dietary and phylogenetic distances. We used (i) Mantel test[54] (with Pearson's correlations, that is, a linear model) and (ii) a non-linear model (Generalized Dissimilarity Modeling (GDM) approach[55]) to measure correlation coefficients between β-diversity and host phylogenetic or dietary distances. All Mantel tests were run with the MRM (Multiple Regression on distance Matrices) function from the 'ecodist' R package[56]. This function has the interesting property of allowing the user to run regression models with multiple predictors, a property that we used to measure the combined effect of host phylogeny and host diet in the prediction of gut community dissimilarities (see below in this section). The Mantel test (with Pearson correlations) assumes a linear model between predictor and response variables. To evaluate the robustness of our results to the potential nonlinearity of the relationship between compositional dissimilarity and host dietary or phylogenetic distances, we also computed correlation coefficients with the GDM approach[55] to relax the linear hypothesis. The use of GDM did not affect the overall correlation patterns and we noted that the explanatory only slightly increases in comparison with the linear Mantel approach (Supplementary Fig. 4).

To test whether $R^2$ values were significantly higher than expected by chance, we used permutation tests on the distance matrices and computed the 95% credibility interval of correlations between community dissimilarities and both host dietary and phylogenetic distances at each phylogenetic slice, producing 95% credibility envelopes for both factors. At each slice, host names were randomly shuffled 100 times in dietary and phylogenetic distance matrices, and correlations with community dissimilarities (β-diversities) were re-computed for each replicate, yielding a distribution of 100 $R^2$ coefficients with respect to each factor. A correlation with either host diet or phylogeny is considered significant when the $R^2$ is higher than the upper bound of the 95% credibility interval.

Finally, variance partitioning[57] was used to quantify the variance in microbiome composition explained by the joint effect of host diet and phylogeny. We first measured the correlation coefficient associated with the union of host phylogenetic and host dietary effects (that is, using both predictors in the same model). Then we computed the correlation coefficient associated with the intersection effect of host phylogeny and diet (equation (3)):

$$\text{Intersection } R^2 = R^2(\text{Phylogeny only}) + R^2(\text{Diet only}) \\ - \text{Union } R^2(\text{Phylogeny} + \text{Diet}) \quad (3)$$

**Distribution and niche of individual bacterial lineages.** At all bacterial scales defined by time or evolutionary distance, we measured the percentage of individual bacterial lineages (OTU) that have a distribution across mammalian hosts correlated to host phylogenetic and dietary distances, using permutation multivariate analysis of variance tests[58] and a false discovery rate (FDR) approach to correct for multiple tests.

The ecological niche of a species (here each bacterial lineage) can be described by its mean and breadth along an environmental gradient. To describe the niche of gut bacterial lineages with respect to host diet, we used the multivariate co-inertia analysis called Outlier Mean Index[59]. This technique enabled us to project both hosts and their diet-associated bacteria onto the same axes and to compute niche means and niche breadth for each bacterial lineage. We used this technique for two reasons. First, contrarily to canonical correspondence analysis or redundancy analysis, the Outlier Mean Index does not assume a particular response curve of the species to the environment (for example, unimodal or linear). Second, this ordination technique gives equal weight to all hosts, independently of their bacterial lineage richness.

**Reconstruction of ancestral communities.** Mammals diverged $> 100$ Myr ago. We expect that saturation of the phylogenetic signal in OTU turnover leads to systematic biases in inference of ancestral community compositions. To overcome these issues, we used the Count program[37] to estimate the ancestral gut community compositions in a phylogenetic and probabilistic framework. Count was originally designed to model the evolution of gene families through gene gain and loss. Along host evolution, the composition of gut microbiota changes through gain or loss of OTUs, so probabilistic models implemented in Count are adequate to model gut microbiota evolution.

OTUs are considered as independent from each other. We used a probabilistic birth–death model of gain and loss of OTUs to compute probabilities of the

presence/absence at each internal node of the mammalian host tree. We used a non-homogeneous version of the gain/loss model where gain and loss parameters are allowed to vary across the branches of the host tree. Variation of gain and loss rates across bacterial lineages is modeled using discrete Gamma distributions, each with four rate categories. All rates were estimated in a ML framework. With all ML rate estimates and the observed presence/absence matrix, we computed the posterior probability for each bacterial lineage (OTU) to be present at each internal node of the host tree. At a given node, the expected total number of OTUs was computed by summing all OTU-specific posterior probabilities of being present at this node.

**Reconstruction of ancestral diets.** We first present methodological details for reconstructing ancestral diets using extant and ancestral microbiome compositions. Our microbiome-based method to predict ancestral diets uses the correlation between extant diet and extant gut bacterial community compositions. Assuming that the relationship between diet and bacterial composition has not varied since the last common ancestor of mammals, ancestral diet can be predicted from reconstructed ancestral community compositions (Fig. 2b). The effects of carnivory and herbivory on the distribution of bacterial lineages are not maximal at the same phylogenetic timescale (Fig. 1d). Thus we selected a time slice (300 Myr ago) allowing us to construct communities having both carnivorous- and herbivorous-specific bacterial lineages to build a regression model to predict diet. We converted the OTU table to presence/absence data and restricted our analysis to bacterial lineages having a distribution across hosts that significantly correlates with diet (permutation multivariate analysis of variance tests, with FDR correction for multiple tests). We then used a principal component analysis (PCA) to project host species according to the composition of their microbiota. Host coordinates along the first axes of the PCA are then used as independent (explanatory) variables in a multinomial logistic regression model, in which the dependent (predicted) variable is diet, discretized into three categories: carnivore, omnivore, and herbivore. The optimal number of independent variables (PCA axes) to include in the regression model can be determined with model selection criteria such as Akaike's Information Criterion (Supplementary Fig. 10a). For each ancestor, its reconstructed community is then projected onto the PCA. Projection coordinates are then used to predict a corresponding diet with the multinomial logistic regression. For each ancestor, the prediction provides a vector of probabilities for each diet category. To visually represent the evolution of diet along a linear gradient from carnivory to omnivory to herbivory, we transformed each vector of diet probabilities into a single value. We assigned 0 to carnivory, 0.5 to omnivory and 1 to herbivory and multiplied each probability by its corresponding weight and then summed the weighted probabilities. The new variable has values between 0 and 1, with 0 representing a probability of 1 to be a carnivorous and 1 a probability of 1 to be herbivorous. We also performed ancestral diet reconstruction at two other time slices to control for the influence of the phylogenetic scale at which community compositions are defined on the ability to accurately infer ancestral diet. The 300 Myr ago time slice defines monophyletic clusters of 16S sequences (OTUs) with, on average, ∼94% similarity. We selected another time slice around 120 Myr ago creating OTUs with ∼97% 16S similarity. The last time slice used to reconstruct ancient diet is the slice at which the correlation between community dissimilarity and diet is maximal (see Fig. 1), around 600 Myr ago, building OTUs with ∼91% 16S similarity.

As for the reconstruction of ancestral diets using extant diets encoded as discrete character traits: we considered diet as a discrete variable with three states (Carnivorous, Omnivorous and Herbivorous). We used the ARD (All-Rates-Different) model of trait evolution implemented in the 'ace' function from the 'ape' R package[60] to estimate the ML transition rates across dietary categories and to infer posterior probabilities for each diet state at each ancestor.

We compared the accuracy and precision of trait- and microbiome-based models as follows. To measure the predictive power of our microbiome-based method, we used cross-validation experiments to evaluate the accuracy of the inference of extant diets using extant microbiome compositions (see Supplementary Fig. 10b,c and 'Accuracy of our microbiome-based method of diet prediction' in Supplementary Note 7). To compute precision of ancestral diets, we compared the trait- and microbiota-based reconstructions on 33 species to the trait-based reconstruction based on >1,500 species because this large taxonomic sampling exhaustively samples mammalian diversity and provides diet inferences that are in agreement with the fossil record[38]. To compute the precision of inferences, we measured the Shannon entropy of each probability vector. The entropy is measured as follows, with $I$ the space of possible diet states and $i \in I$:

$$\text{Entropy} = -\sum_i p_i \log p_i \qquad (2)$$

The entropy measures how spread a probability distribution is: if the probability vector is uniform, the entropy is maximal. However, if the distribution is concentrated on a single category (meaning that probabilities are low for other categories), the entropy is low. In our case, as we are making predictions, we want to obtain probability distributions with entropies as low as possible, to be able to assign with confidence an ancestral diet to a given ancestor.

**Phylosymbiosis along the host phylogeny.** To measure phylosymbiosis, we used gut microbiota compositions defined at a recent phylogenetic timescale, recent enough so that the correlation with host phylogeny is near maximal but deep enough in the phylogeny to ensure that bacterial lineages appear in a sufficient number of hosts. The selected slice creates 1,484 bacterial lineages (OTUs) that are observed in ≥2 hosts. On average, each OTU has an observed DNA similarity of ∼97%. We rarefied the OTU table to avoid reconstruction biases owing to unequal sequencing across samples.

In its original definition[4,5,61], phylosymbiosis has been described as the congruence between the host tree and the tree of communities, inferred using the compositional dissimilarities between gut microbiota. Previous attempts to detect and measure phylosymbiosis from gut microbiome data employed parsimony- and distance-based methods to compute the tree of communities (for example, Neighbor Joining[62]), without probabilistic modelling of OTU evolution. In the case of parsimony[4,11], previous authors considered all OTUs independently and recoded OTU relative frequencies into ordered discrete states, from 0 to 6, to reflect log-unit differences in their occurrence. The tree minimizing the number of transitions between these ordered states was considered as the Maximum Parsimony tree of communities. For distance-based methods, a dissimilarity matrix computed with a β-diversity metric, such as Bray–Curtis or UniFrac, was used to compute a dendogram (that is, the tree of community dissimilarities) with a hierarchical clustering method[5,61]. However, over millions of years of host evolution, many OTU gain and loss events have occurred, possibly randomizing the genuine historical signal of compositional change along the mammalian phylogeny. In molecular phylogenetics, parsimony- and distance-based approaches are known to reconstruct less accurate topologies and to be more sensitive to substitution saturation than probabilistic approaches that employ evolutionary substitution models[63,64]. Here, at the scale of mammals, which diverged >100 Myr ago, we also expect issues related to the saturation of the phylogenetic signal in OTU turnover, leading to systematic biases in community tree reconstruction. Using the mammalian data set that we re-analysed in this study, Muegge et al.[11] used approaches that did not employ explicit modelling of OTU dynamics of gain and loss to search for patterns of phylosymbiosis. They concluded that mammalian community compositional dissimilarities did not mirror host distances, questioning the existence of long-term phylosymbiosis in mammals.

Here we extend the original definition to apply the 'phylosymbiosis' concept to each individual host clade. Using ancestral community reconstruction with a model-based and probabilistic approach, we measured for each ancestor the amount of bacterial lineages that were ancestrally present and compared this number to a null expectation in which the observed relationships between hosts and bacteria are disrupted. To compute this null expectation, we ran Count under the same non-homogeneous gain/loss model (see 'Reconstruction of ancestral communities' in the Methods), with 100 random OTU tables created with the independent swap algorithm[65] (50,000 iterations). This algorithm maintains OTU occurrence frequency across hosts and sample-specific richness during the randomization of the data and is widely used in ecology[65]. With these random distributions, we computed $P$ values for each ancestor to detect those that have a significantly higher number of present OTUs than randomly expected; these ancestors then define clades with mammalian species that possess gut microbiomes that share more bacterial lineages between each other than randomly expected. We computed the magnitude of the phylosymbiosis signal along the mammalian host tree, for each ancestor, by computing a SES per branch/node (see Results, equation (1)).

After measuring the magnitude of the phylosymbiosis signal, we investigated its pattern along the phylogeny of hosts. First, we tested whether the time span of microbiome turnover over millions of years of host evolution has an impact on phylosymbiosis, yielding to weaker phylosymbiosis signals for the most ancient mammalian clades and higher signals for the most recently diverged clades. We correlated the amount of phylosymbiosis (SES values) to crown ages of each focal mammalian ancestor (Fig. 3). Although we computed a $R^2$ coefficient, we did not test whether the correlation was significant, because the data points are non-independent. Second, we tested whether dietary shifts during mammal evolution reduced the amount of bacterial lineages shared within mammalian clades and negatively impacted the signal of phylosymbiosis. We correlated the residuals of the linear regression between the amount of phylosymbiosis and time (SES–age) to a quantification of dietary shifts. To measure these dietary shifts, we computed dietary distances between successive nodes in the mammalian phylogeny. We used ancestral diets reconstructed both with the trait-based approach using the dietary data of 1,534 mammals and with the microbiome-based approach. Each approach provides a probability distribution for each dietary category at each ancestral mammalian node. We computed the dietary distance between two consecutive nodes by computing a distance between the two probability distributions. When computing this distance, we accounted for the fact that omnivory is the union of carnivory and herbivory. Each node has three probabilities (summing to one) that correspond to the three diets (omnivory, carnivory and herbivory). We first assigned one diet (omnivory, carnivory or herbivory) to each node by drawing a diet category from a multinomial distribution defined with the three probabilities. Second, we recoded the diet as a two state variable, either 'Carnivore' or 'Herbivore': if the node was first assigned 'Omnivore', the node is now assigned both Carnivore and Herbivore. Three configurations are possible: herbivore only, carnivore only, or both herbivore and carnivore. Third, we computed dietary

distances between two nodes using the Sørensen metric. This procedure[66] enables us to account for the fact that a carnivorous (or herbivorous) diet is a subset of an omnivorous diet. We repeated this procedure 100 times (draw samples from the multinomial distribution differ at each repetition) and we used the mean dietary distance between the two distributions.

**Detecting co-speciating bacterial lineages.** We measured the amount of co-speciation and host-swap events per OTU at the same time slice that we used to measure the phylosymbiosis signal. We only conserved OTUs that are present in at least four hosts and we used the rarefied data matrix. We extracted the individual alignments for each OTU from the global alignment previously used to reconstruct the phylogenetic tree of all 16S sequences. Replicate sequences were treated as follows. When identical sequences were observed multiple times in a single host, only one sequence was retained in the alignment. Replicate sequences observed in multiple hosts were all conserved in the OTU alignment.

We used the ALE algorithm[39–41] to detect the co-speciation and host-shift events along the phylogenetic tree of 16S sequences. ALE employs a probabilistic and event-based approach, reconciling the symbiont phylogenetic tree with the mammalian host tree using a probabilistic model of co-speciation, host-swap, intrahost speciation and extinction of symbionts within the host tree. Initially, the ALE algorithm has been designed to reconcile gene trees with species trees with a model of gene speciation, transfer, duplication and loss. In the case of reconciliation between a symbiont tree and a host tree, these events correspond to co-speciation, host-swap, intrahost speciation and extinction events, respectively. ALE has three key features: (i) it is implemented in a probabilistic framework, (ii) it makes use of time calibrations in the host tree to constrain in time the host swaps during symbiont evolution (host swaps cannot happen back in time), and (iii) it also accounts for uncertainties in the OTU tree reconstruction. In the following, only the relative amount of co-speciation versus host swap is considered, as these events allow us to measure the amount of vertical versus horizontal inheritance in the gut microbiota.

ALE uses the information in the alignment of sequences to search for the ML values of co-speciation and host-swap rates. When the associated ML symbiont tree has been found, ALE provides the ML reconciliation scenario including the numbers of co-speciation and host swap. ALE uses a distribution of symbiont trees to search for the best reconciliation scenario. We run Phylobayes[67] to obtain a posterior distribution of OTU trees with a GTR model and a discrete gamma distribution with four categories to model rate variation across sites. We run Phylobayes for 10,000 generations, sampling at every generation with an initial burn-in of 1,000 generations.

We then compared the amount of OTUs that harbour more co-speciation than host swap with a null expectation. This null model allowed us to measure the amount of OTUs that were observed to have more co-speciations than host swaps owing to chance or to the overfitting to the host tree, during ALE reconciliations. We proceeded as follows. We shuffled the names of the host tree to break up the initial host phylogenetic relationships and re-ran ALE to measure the amount of co-speciation and host swaps under this null expectation. We performed 100 replicates of the null model per OTU. For each OTU, we computed the number $N_{null}$ of null replicates for which the amount of co-speciation events is equal to or higher than the number of host-swap events. We computed a $P$ value for each OTU by dividing $N_{null}$ by the number of replicates $+1$. The null expectation of the number of OTUs harbouring more co-speciation than host-swap events is then simply obtained by summing the $P$ values (null expectations are represented in Fig. 4a, 'Null expectation' bars).

To control for overfitting, we computed for each replicate $r$ the difference $D_{r,null}$ between the number of co-speciations and the number of host swap and compared this null difference with the observed difference $D_{r,obs}$ for this OTU. We computed a $P$ value for each OTU, defined as the number of times $D_{r,null}$ is $\geq D_{r,obs}$, divided by the number of replicates $+1$. After correcting for multiple tests with an FDR approach, if a $P$ value is higher than the significance threshold (here, 0.05), then we cannot reject the hypothesis that chance and/or overfitting to the host tree are the reasons for observing more co-speciations than host swaps. The proportion of OTUs that were observed to have more co-speciation than host swap owing to overfitting is represented in grey in Fig. 3c.

**Code availability.** BDTT has been implemented in R. All codes needed to run BDTT, as well as illustrative examples, are available here: https://github.com/FloMazel/BDTT.

**Data availability.** The data that support the findings of this study (the multiple sequence alignment of the processed 16S data that we used for all phylogenetic analyses, the OTU table of unique sequences, the calibrated and non-calibrated bacterial phylogenetic trees used in the BDTT analyses, the matrix of host phylogenetic distances and the matrix of host dietary distances) are available from: https://github.com/mgroussi/MammalianGuts

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

## Acknowledgements

This work was funded by The Center for Microbiome Informatics and Therapeutics and is also supported by the National Science Foundation under Grant No. 0821391. F.M., S.L. and W.T. received support from the European Research Council under the European Community's Seven Framework Programme FP7/2007-2013 Grant Agreement no. 281422 (TEEMBIO). F.M., S.L. and W.T. belong to the Laboratoire d'Écologie Alpine, which is part of Labex OSUG@2020 (ANR10 LABX56). Part of the computations was performed using the CiGRI platform of the CIMENT infrastructure (https://ciment.ujf-grenoble.fr), which is supported by the Rhône-Alpes region (GRANT CPER07_13 CIRA) and France-Grille (http://www.france-grilles.fr). We are thankful to Laure Gallien for her guidance in the use of the VirtualCom R package.

## Author contributions

M.G., F.M. and E.J.A. conceived and designed the study. M.G. and F.M. performed all analyses. M.G., F.M., C.S.S., J.G.S. and E.J.A. analysed the data. J.G.S., C.S.S., S.L. and W.T. contributed to the writing of the manuscript. M.G., F.M. and E.J.A. prepared the manuscript.

## Additional information

**Competing financial interests:** The authors declare no competing financial interests.

