## [Peer Review File · Nature Communications]

Reviewers' comments:

Reviewer #3 (Remarks to the Author):

This is an interesting manuscript on a challenging topic that unfortunately still needs substantial improvement. I do have a couple of new ideas for how to achieve this – see major points 1 and 2.

Major point 1

I think to make a fair comparison between “diet” and “phylogeny” the authors need to create random variables that change on the host phylogeny at the same rate as diet does. Then the predictiveness of these random variables can be compared directly with that of diet. Apples would be being compared with apples.

Phylogenetic distance is bound to be more predictive at the tips of the tree and less predictive at the top than either diet OR any of these random variables, simply due to it being extremely fine grained metric. The only part of figure 1 that I find meaningful is the comparison between bacteria associated with herbivory and carnivory.

Major point 2

It is not obvious to me how much the first half and the second half of the manuscript really have to do with each other. Quite possibly, it would be better split into two with supplementary text integrated into the main text. Two clear focussed manuscripts would be much better than what we have currently, which is a marathon for the reader. I think currently it is simply too ambitious in trying to build a synthesis and falls down in lots of different ways but most especially in terms of presentation. Statistical tests to compare effects of diet and vertical inheritance are welcome and indeed could fit into either manuscript but most of those currently presented are not very convincing...

The two manuscripts would be:

- (1) How strong is the polysymbiosis signal in mammals?
- (2) Effect of diet on mammalian microbiota.

I suppose the bottom line is I just do not accept the "disentangled".

Major point 3

I still find this manuscript difficult to understand, due to language which is abstract and imprecise. For example, there are about 101 ways the sentences below from the abstract

can be interpreted. I think I know approximately the correct ones but this is based on having read and thought about the manuscript and even then the residual uncertainties make the substance of the overall claims hard to evaluate:

Here, we show that host phylogeny and diet, despite being deeply confounded, select non-overlapping gut bacterial lineages, and do so on vastly different timescales.

Host diet is something entirely concrete, what the host eats. Host phylogeny means the position on the phylogeny of the hosts relative to the others. I take it that deeply confounded means simply that differences in diet evolve slowly and therefore do not change many times in the phylogeny.

Non overlapping is a very strong claim. To say that no-diet selected lineages also show any correlation with phylogeny is very strong, if this is the claim being made. I think what is meant is that the lineages that show the strongest (or detectable, given the dataset you have) signals of phylogenetic correlation are different from the ones that show the strongest (or detectable, given the dataset you have) dietary correlation. But either of these weak versions of the claim would be unsurprising. The strong claim is implausible. Given a large enough dataset, it seems likely that essentially any bacteria would show some phylogenetic correlation and the vast majority would also show some dietary correlation as well.

"different timescales" is not referring to the speed of evolution, nor the selection coefficient but in fact bacterial phylogenetic distance. I only know that because I read the manuscript. Throughout it is not clear enough whether time really means time or whether it is bacterial distances/times or host distances/times that are being referred to.

However, associations with host phylogeny are mostly seen among more recent lineages, driven by a process operating at the same time scale as host evolution.

Associations of what, exactly? I think you mean bacterial phylogeny but exactly what is not clear. I may be missing something but it seems to me that associations of anything with host phylogeny have to be at least in some senses driven on the same time scale as host evolution. And I believe that "more recent lineages" are groups of related bacteria that diverged recently but only because I read the manuscript.

More detailed phylogenetic analyses support co-speciation as playing a significant role in the evolution of mammalian gut symbionts.

In what sense more detailed? And more detailed than what, exactly. Also, I think all that you mean is there is a signal of cospeciation, not that cospeciation itself has had evolutionary consequences. This would be very interesting of course but it is a potential implication of your current findings, rather than a finding.

Diet mostly influences the acquisition of deeply divergent microbial lineages. This sentence does not actually make sense, literally it implies that diet causes the

acquisition of bacteria that are dissimilar from each other. Diet influences some bacteria more than others. What I think you mean is that groups of bacteria that are strongly predicted by particular diets form large (and therefore old) clumps on the tree of life.

The introduction is rather clearer than the abstract in terms of language. However by the below:

We hypothesized that these two factors may have driven vertical and horizontal inheritance of bacterial lineages at different phylogenetic scales.

It's unclear whether you mean host phylogenetic scales or bacterial phylogenetic scales. I think it is probably host. Host switches happen on a phylogeny. They do not happen at "phylogenetic scales", they happen at specific places on the host tree. It's far too unclear what you actually mean.

And by the below

However, if vertical inheritance is not involved, associations with host phylogeny should be seen at timescales of bacterial evolution that are decoupled from host evolution.

I think this is what you actually mean is:

However, if vertical inheritance is not involved, associations with host diet would be seen at a variety of timescales of bacterial evolution, reflecting the rate of adaptation of bacteria to specific diets. Correlations between bacteria and host will be driven by the correlation of host diet with host phylogeny but the bacterial lineages involved can be either much younger or much older than the set of related hosts that share the same diet.

i.e. If vertical inheritance does not matter, then host phylogeny will only predict bacteria insofar as it predicts diet.

Note (somewhat tangentially) that if a particular diet only evolved once, then the bacterial lineages that evolved to take advantage of that diet might well be the same age as the diet itself. So in that case, they may not be uncoupled temporally.

while the correlation with diet would be primarily driven by horizontal inheritance, if diet changes slowly, this is not necessarily clear.

And then at the end in the final sentence of the introduction, you seem to in some way equate host phylogeny with vertical and diet with horizontal, which seems terribly unhelpful.

may allow us to disentangle the individual contributions of host phylogeny and diet, and to understand how and to what extent these vertical and horizontal inheritances have driven gut community evolution.

I continue to think it is false to say that the effects of diet and phylogeny are being disentangled. The point is that things that are correlated with diet are still vertically inherited through large parts of the phylogeny. Finding different patterns to statistical

correlations is different from disentangling, which would entail e.g. explicitly detecting bacterial switches due to dietary switches.

I do not see at all what it proves that the lineages that are not associated with diet have correlations with host phylogeny that are nearly as strong as all of them. Is there a way of directly comparing diet and non-diet associated lineages of a similar age? In any case, the claims about "non-overlapping" seem to come out of thin air and indeed seem to be contradicted by this sentence:

some bacteria related to host phylogeny at recent time scales are nested within higher clades also related to host diet,

Figure 1D, not clear enough at least based on main text/figure legend what was done and what is being shown.

The reasoning that there are no omnivory associated taxa is not clear enough in the main text/figure.

The main text of the latter part of the results should be integrated with the supplementary text, as it is extremely hard to get anything out of it as to what is actually going on.

Reviewer #4 (Remarks to the Author):

This is an interesting paper by Groussin and colleagues, investigating the gut microbiome of 33 mammal species using a new methodology. I identified several issues:

1. According to today's standards, the 16S sequencing dataset used here (from Muegge et al.) is low quality. There is a small number of reads (seems like around 44,000 total from 33 samples). In principle, as a reviewer, I am not in favor of asking authors to collect and generate more data. Nevertheless, these issues greatly reduce my confidence in the validity of the results, and I would recommend that the authors address this issue explicitly in the text, as well as include analyses that show that the below-standard dataset is not influencing the result. Maybe some sort of a subsampling analysis, simulations, or other (newer) publicly available datasets used as a point of comparison, to show the results using this small dataset are unbiased?

2. Another problematic issue is that only a single individual is sampled from each species. Thus, this analysis cannot account for within-species variation, which could have a large effect, as we know from human studies. As above, I would encourage the authors to address this and alleviate these concerns. Maybe using other available datasets that contain multiple individuals from each species to show the selection of a single individual from each species doesn't bias the result?

3. I might have missed it, but I could not find a link to view or download the data used here. These should be made freely available to reviewers and readers who may be interested in replicating the study or re-analyzing the data for new biological findings. Ideally, there should be a link to download all the processed datasets used in the analysis.

**Reviewers' comments:**

**Reviewer #3 (Remarks to the Author):**

**This is an interesting manuscript on a challenging topic that unfortunately still**
**needs substantial improvement. I do have a couple of new ideas for how to achieve**
**this – see major points 1 and 2.**

**Major point 1**

**I think to make a fair comparison between “diet” and “phylogeny” the authors need**
**to create random variables that change on the host phylogeny at the same rate as**
**diet does. Then the predictiveness of these random variables can be compared**
**directly with that of diet. Apples would be being compared with apples.**

This is a good suggestion for a control that we did not think of in the previous draft. We
have performed simulation experiments as suggested by the reviewer. We have estimated
the transition rates between dietary states (herbivory, carnivory and omnivory) with the
ARD (All Rates Different) Markovian model (implemented in the ape R package) along
the phylogeny of 1,534 mammals that we have used elsewhere in the paper (note that the
ARD model was selected because it is the model that best fits the data among all models
that we have tested). We used the ML estimates of these transition rates to simulate traits
along our phylogeny of 33 mammals (100 replicates), so that each trait is forced to evolve
at the same rate as diet does along the host phylogeny. Then we computed trait distance
matrices and performed a BDTT analysis to compare the explanatory power (R^2) of these
simulated traits to the one of the observed diet. Supp. Fig. 8 (below) shows that the
simulated traits poorly predict the compositional dissimilarities of our mammalian gut
microbiomes. Importantly, **we do not observe any increase in explanatory power**
**when computing correlations at ancient time scales**, ruling out the possibility that the
peak of correlation with observed diet at ancient time scales is only driven by the coarse
granularity of the dietary distance matrix. It also further supports the claim that there is an
effect of diet that is independent from the host phylogeny.

Supp. Fig. 8: The peak in correlation between diet and gut microbiome compositions at
ancient timescales is not simply an echo of phylogenetic history written in diet.

We simulated phylogenetically-conserved traits that evolved along the mammalian
phylogeny at the same rate as diet does, and compared the correlation profiles between
these simulated traits and microbiome compositions with the correlation profile obtained
with observed diets. The distributions of simulated correlation profiles are represented in
the form of a 95% null envelope. The dark red plain line connects the medians of these
distributions. The dark red dashed line connects the 95% quantiles. The original
correlation profile with observed diets is in orange and is the same as in Fig. 1A. The
high correlation with observed diets at ancient timescales is significantly higher than the
null, showing that there is a genuine signal associated with diet that is independent from
the host phylogenetic history written in diet.

**Phylogenetic distance is bound to be more predictive at the tips of the tree and less**
**predictive at the top than either diet OR any of these random variables, simply due**
**to it being extremely fine grained metric. The only part of figure 1 that I find**
**meaningful is the comparison between bacteria associated with herbivory and**
**carnivory.**

Thank you for this comment. The hypothesis formulated by the reviewer is that the fine
granularity of the host phylogenetic distance matrix (compared to the coarse granularity
of the diet distance matrix) is biasing the sections of the bacterial tree where the
predictive power for microbiome compositions is high towards the tips of the tree. If this

is true, using coarse-grained host phylogenetic distance matrices should displace the area
where the correlation with host phylogeny is maximum towards more ancient regions of
the bacterial tree, just as we observe for diet. We have tested for this possible bias as
follows. We re-ran BDTT using a series of coarse-grained distance matrices for host
phylogeny. We reasoned that if the fine-grained distances were leading to the peak at
more recent times, then using more coarse-grained distances for the host would lead to a
correlation at older distances. To test this hypothesis, we used a set of thresholds to
define new, more coarse-grained host phylogenetic distances matrices, with all pairwise
distances below these thresholds set to null. Supp. Fig. 7 (below) shows that when
coarse-grained host phylogenetic distance matrices are used, the correlation with host
phylogeny is always localized at recent time scales on the bacterial phylogeny, separated
from the highest correlations with host diet along the phylogeny of bacteria. In fact, when
the most coarse-grained host distance matrix is used, the signal disappears entirely, and
never shifts toward more ancient times. This control experiment, along with our new
simulation experiment detailed above, further confirms that host phylogeny and diet
impact gut microbiome compositions at different bacterial phylogenetic scales, and that
these effects can be partitioned with our BDTT approach, which is illustrated in Figure 1.

Supp. Fig. 7: The high correlation with host phylogeny in recent regions of the bacterial tree does not depend on the high-granularity of the matrix of host phylogenetic distances. The top left panel shows the distribution of all pairwise host distances in time units between our 33 mammals. The other panels are replicates of Fig. 1A, using different granularities for the matrix of host phylogenetic distances, from fine-grained (top right panel) to coarse-grained (bottom panels) matrices. PHPD: Pairwise Host Phylogenetic Distance. For a given plot, all PHPDs below a given distance threshold are set to 0, decreasing the granularity of the original distance matrix. When the granularity of the host phylogenetic distance matrix is getting coarse, the correlation with gut microbiome compositions is decreasing, as expected. However, the maximum of this correlation is not shifting towards more ancient regions of the bacterial tree, and the scale disparity between the effects of host phylogeny and diet is still observed.

Major point 2

It is not obvious to me how much the first half and the second half of the manuscript really have to do with each other. Quite possibly, it would be better split into two with supplementary text integrated into the main text. Two clear focussed manuscripts would be much better than what we have currently, which is a marathon for the reader. I think currently it is simply too ambitious in trying to build a synthesis and falls down in lots of different ways but most especially in terms of presentation. Statistical tests to compare effects of diet and vertical inheritance are welcome and indeed could fit into either manuscript but most of those currently presented are not very convincing...

The two manuscripts would be:

- (1) How strong is the polysymbiosis signal in mammals?**
- (2) Effect of diet on mammalian microbiota.**

We thank the reviewer for this suggestion. We agree that that there is a lot of material in this manuscript. At the same time, there is some advantage to presenting them together in a single paper to show how the different processes involved in shaping gut microbiome compositions can be modeled and quantified in future evolutionary analyses of host-associated microbiome data. The first half of our manuscript also provides necessary context for the analyses presented in the second half. Given the arguments to be made for and against splitting the manuscript, we will seek guidance from the editor on the most appropriate way to present the work at *Nature Communications*.

I suppose the bottom line is I just do not accept the "disentangled".

Regarding the disentangling of host phylogeny and diet: we agree that it is a very difficult task, and our approach is certainly not perfect. Indeed, for traits evolved on the same phylogeny, it simply may not be possible to completely disentangle their effects. However, we think that the collection of all of these experiments and results represent a significant improvement from what has been presented in the literature in the past.

That said, we have added a sentence in the manuscript to explicitly state that we are only partitioning the main effects of host phylogeny and diet, and that it might not be possible to entirely disentangle the factors themselves ("Note that BDTT allows us to statistically disentangle the temporally separated portions of the contributions of host phylogeny and diet (when defined with a coarse granularity), not the totality of the processes themselves.", Lines 133-136). We again explicitly discuss these issues later in the text when presenting the results on co-speciation (lines 321-328).

In addition, we have changed the title, removing the reference to disentangling diet and
phylogeny. The new title is: “Unraveling the processes shaping mammalian gut
microbiomes over evolutionary time”

In addition, we now clearly state in the text that the effects of diet that we can capture are
only those that are subsequent of major dietary shifts, which are uncorrelated to host
phylogeny at the scale of all mammals. Furthermore, our study gives researchers a more
concrete way to assay whether (host-)phylogenetically-correlated signals in microbiome
data are likely to be driven by contemporaneous coevolution — a question about which
there is frequently some confusion. Finally, while host phylogeny and host diet are often
considered as ‘competing’ explanatory factors in the literature (Carmody et al., 2015 Cell
Host & Microbe), we show that they actually act at different (bacterial phylogenetic)
scales.

Of course, we are aware that the evolutionary trajectory of some bacteria might be driven
by dietary differences that are themselves correlated to host phylogeny, and that it is
difficult for us to disentangle the effects of host phylogeny and diet in these cases.
However, there has been little attempt to partition their main effects at this phylogenetic
scale in mammals, which has made it difficult for the research community to develop an
intuition for the individual effect of both factors. Within the bounds of what we can
actually achieve with these kinds of data, we think that our manuscript provides an
interesting dissection of the main effects of host phylogeny and diet.

**Major point 3**

**I still find this manuscript difficult to understand, due to language which is abstract**
**and imprecise. For example, there are about 101 ways the sentences below from the**
**abstract can be interpreted. I think I know approximately the correct ones but this**
**is based on having read and thought about the manuscript and even then the**
**residual uncertainties make the substance of the overall claims hard to evaluate:**

**Here, we show that host phylogeny and diet, despite being deeply**
**confounded, select non-overlapping gut bacterial lineages, and do so on vastly**
**different timescales.**

**Host diet is something entirely concrete, what the host eats. Host phylogeny means**
**the position on the phylogeny of the hosts relative to the others. I take it that deeply**
**confounded means simply that differences in diet evolve slowly and therefore do not**
**change many times in the phylogeny.**

We have attempted to make changes in the abstract to clarify all these points that are
mentioned given word limits (see below). In particular, we have removed the term “non-
overlapping” from the text, and rephrased the abstract and the paragraph in the main text
presenting these results (lines 137-154).

**Non overlapping is a very strong claim. To say that no-diet selected lineages also**
**show any correlation with phylogeny is very strong, if this is the claim being made. I**
**think what is meant is that the lineages that show the strongest (or detectable, given**
**the dataset you have) signals of phylogenetic correlation are different from the ones**
**that show the strongest (or detectable, given the dataset you have) dietary**
**correlation. But either of these weak versions of the claim would be unsurprising.**
**The strong claim is implausible. Given a large enough dataset, it seems likely that**
**essentially any bacteria would show some phylogenetic correlation and the vast**
**majority would also show some dietary correlation as well.**

Concerning the “non-overlapping” claim — as said above, we have removed this term
from the paper. We agree that this claim depends on the data that we have. That said,
these data are representative of the part of the microbiome containing the most abundant
bacterial taxa in each of these mammals. Among these bacterial lineages, we clearly
observe that some bacterial lineages that show correlation with host phylogeny do not
exhibit distributions across hosts that correlate with diet, and vice versa. For instance,
*Bacteroides fragilis* has developed host-specific interaction mechanisms with the host
epithelium cells, allowing it to colonize the gut of all mammals, irrespective of diet (e.g.
Lee et al., Nature, 2013). Members of the fiber-degrading Prevotellaceae family are
frequently observed in our plant-eating mammals, irrespective of their phylogenetic
distances. Finally, Moeller et al. (Science, 2016) have shown that even at the short time
scale of Hominid evolution, spore-former Lachnospiraceae bacteria do not harbor co-
speciation patterns with host phylogeny, highlighting the fact that some bacteria can
colonize hosts without any specificity regarding their phylogenetic distances.

**“different timescales” is not referring to the speed of evolution, nor the selection**
**coefficient but in fact bacterial phylogenetic distance. I only know that because I**
**read the manuscript. Throughout it is not clear enough whether time really means**
**time or whether it is bacterial distances/times or host distances/times that are being**
**referred to.**

Thank you for this comment. We have modified the abstract, which now states “[...] and
do so on vastly different *bacterial evolutionary* timescales”

**However, associations with host phylogeny are mostly seen among more recent**
**lineages, driven by a process operating at the same time scale as host evolution.**

**Associations of what, exactly? I think you mean bacterial phylogeny but exactly**
**what is not clear. I may be missing something but it seems to me that associations of**
**anything with host phylogeny have to be at least in some senses driven on the same**
**time scale as host evolution. And I believe that “more recent lineages” are groups of**
**related bacteria that diverged recently but only because I read the manuscript.**

We have rephrased this section. We now state:

“Conversely, correlation with host phylogeny is mostly seen among more recently-
diverged bacterial lineages”

**More detailed phylogenetic analyses support co-speciation as playing a significant**
**role in the evolution of mammalian gut symbionts.**

**In what sense more detailed? And more detailed than what, exactly. Also, I think all**
**that you mean is there is a signal of cospeciation, not that cospeciation itself has had**
**evolutionary consequences. This would be very interesting of course but it is a**
**potential implication of your current findings, rather than a finding.**

Thank you for this comment. We have modified this statement to be more accurate:

“Phylogenetic analyses support co-speciation as playing a significant role in the evolution
of mammalian gut microbiome compositions.”

**Diet mostly influences the acquisition of deeply divergent microbial lineages.**
**This sentence does not actually make sense, literally it implies that diet causes the**
**acquisition of bacteria that are dissimilar from each other. Diet influences some**
**bacteria more than others. What I think you mean is that groups of bacteria that**
**are strongly predicted by particular diets form large (and therefore old) clumps on**
**the tree of life.**

You are right, this is what we mean. We now state:

“Diet mostly influences the acquisition of ancient microbial lineages.”

**The introduction is rather clearer than the abstract in terms of language. However**
**by the below:**

**We hypothesized that these two factors may have driven vertical and horizontal**
**inheritance of bacterial lineages at different phylogenetic scales.**

**Its unclear whether you mean host phylogenetic scales or bacterial phylogenetic**
**scales. I think it is probably host. Host switches happen on a phylogeny. They do not**
**happen at “phylogenetic scales”, they happen at specific places on the host tree. Its**
**far to unclear what you actually mean.**

We are sorry for this confusion. We actually meant “bacterial” phylogenetic scales. We
have added this detail (Line 44).

**And by the below**

**However, if vertical inheritance is not involved, associations with host phylogeny**
**should be seen at timescales of bacterial evolution that are decoupled from host**
**evolution.**

I think this is what you actually mean is:

However, if vertical inheritance is not involved, associations with host diet would seen at a variety of timescales of bacterial evolution, reflecting the rate of adaptation of bacteria to specific diets. Correlations between bacteria and host will be driven by the correlation of host diet with host phylogeny but the bacterial lineages involved can be either much younger or much older than the set of related hosts that share the same diet.

i.e. If vertical inheritance does not matter, then host phylogeny will only predict bacteria insofar as it predicts diet.

Note (somewhat tangentially) that if a particular diet only evolved once, then the bacterial lineages that evolved to take advantage of that diet might well be the same age as the diet itself. So in that case, they may not be uncoupled temporally.

As explained above, there is a clear effect of the ancient major dietary shifts that is independent of the effect of host phylogeny. Host phylogeny can actually encompass multiple factors that are perfectly correlated to host phylogenetic distances, such as genetic, physiological or historical factors, which can all impact gut microbiome compositions irrespective of the influence of diet (most notably genes involved in dialogues between bacteria and the immune system). These host phylogeny-related traits can select for bacterial lineages that are either older, younger or contemporary with the set of hosts that share these traits.

The sentence quoted above (“However, if vertical inheritance is not involved [...] decoupled from host evolution”) actually only concerns these traits that shape microbiome composition independently from diet. We have rephrased the whole section to provide more explanations and background to the reader (lines 43-57).

while the correlation with diet would be primarily driven by horizontal inheritance,

If diet changes slowly, this is not necessarily clear.

Absolutely, you are correct. Fine-scale differences in diet might be correlated with host phylogeny, leaving patterns that are not distinguishable from the effect of host phylogeny (as explained above, we discuss these notions in lines 321-328 and Supplementary Discussion section 2.10). However, our paper is only focusing on the effect of large differences in diet on the composition of microbiomes (as explained in lines 35-40). And these large shifts occurred frequently in the history of mammals (see Price et al, 2010, PNAS). At the scale of our 33 mammals, we observe that when defining diet using large categories (herbivory, omnivory, carnivory), 23% of mammalian lineages experienced switches of diet (15 out of 64 branches). As shown in Supp. Fig. 11, these shifts are congruent with horizontal (and parallel) acquisitions of diet-related bacterial lineages.

And then at the end in the final sentence of the introduction, you seem to in some

400 way equate host phylogeny with vertical and diet with horizontal, which seems
terribly unhelpful.
may allow us to disentangle the individual contributions of host phylogeny and diet,
and to understand how and to what extent these vertical and horizontal inheritances
have driven gut community evolution.

Thank you for this comment. We agree with this point. We have rephrased this section to
be clearer about our hypotheses (lines 58-62). We now state that major dietary shifts were
associated with horizontal inheritance of diet-specific bacteria. Host phylogeny, however,
can be associated with both horizontal and vertical inheritance, as explained in Supp. Fig.
1. Our paper provides evidence of these horizontal acquisitions when shifting between
main dietary categories (Supp. Fig. 11) and provides a quantitative measurement of the
part of the correlation signal with host phylogeny that is congruent with vertical
inheritance of bacterial lineages (Figure 4).

**I continue to think it is false to say that the effects of diet and phylogeny are being**
**disentangled. The point is that things that are correlated with diet are still vertically**
**inherited through large parts of the phylogeny. Finding different patterns to**
**statistical correlations is different from disentangling, which would entail e.g.**
**explicitly detecting bacterial switches due to dietary switches.**

**I do not see at all what it proves that the lineages that are not associated with diet**
**have correlations with host phylogeny that are nearly as strong as all of them. Is**
**there a way of directly comparing diet and non-diet associated lineages of a similar**
**age? In any case, the claims about “non-overlapping” seem to come out of thin air**
**and indeed seem to be contradicted by this sentence:**

**some bacteria related to host phylogeny at recent time scales are nested within**
**higher clades also related to host diet,**

Within the bounds of what we can do with these data, we clearly observe that a vast
majority of the bacterial lineages that have a distribution across hosts that is correlated
with host phylogenetic distances are not nested within larger bacterial clades that are
correlated with coarse-grained diet (*i.e.* main dietary categories). Only a small fraction of
those do show nestedness patterns, which we explain with this sentence (“some bacteria
related to host phylogeny at recent time scales are nested within higher clades also related
to host diet”, line 144-147 in the main text). As explained above in this response (lines
206-243 and 278-291), we know that some bacterial lineages are strongly expected to be
linked to host phylogeny independently of diet, and others to be influenced by diet,
independently of host phylogeny. Our results confirm these expectations at the scale of
the microbiome. Finally, when measuring the effects of diet, we define diet with a coarse
granularity and are only focusing on the impact of large dietary differences on the gut
microbiome composition. Our results show that these effects can be reasonably
partitioned from those of factors that are more intimately correlated to host phylogeny,
including, of course, small differences in diet.

**Figure 1D, not clear enough at least based on main text/figure legend what was done**
**and what is being shown.**

We thank the reviewer for this. We have added details in the legend to clarify what was
done for Figure 1D.

**The reasoning that there are no omnivory associated taxa is not clear enough in the**
**main text/figure.**

Thank you for this comment; we have edited the legend to make it clearer.

**The main text of the latter part of the results should be integrated with the**
**supplementary text, as it is it is extremely hard to get anything out of it as to what is**
**actually going on.**

We agree that this latter part is more speculative than the rest of the study, and we
explicitly state that future studies are needed to confirm our last results. But this section is
important because it provides some of the first (albeit merely suggestive) evidence that
some of the bacterial lineages that are putatively co-speciating with mammals and that
are present in humans are functionally linked to human health. We feel that this analysis
provides a compelling and broadly accessible connection between the largely theoretical
arguments earlier in the manuscript, and issues that are likely to be closer to the research
interests of many in the readership of *Nature Communications*. Showing that the more
tightly co-speciating bacterial lineages present in humans are also enriched among the
genera that were found to be negatively associated with IBD coherently extends the
previous section on co-speciation patterns at the scale of all mammals and suggests some
more mechanistic hypotheses for future study.

**Reviewer #4 (Remarks to the Author):**

**This is an interesting paper by Groussin and colleagues, investigating the gut**
**microbiome of 33 mammal species using a new methodology. I identified several**
**issues:**

**1. According to today's standards, the 16S sequencing dataset used here (from**
**Muegge et al.) is low quality. There is a small number of reads (seems like around**
**44,000 total from 33 samples). In principle, as a reviewer, I am not in favor of asking**
**authors to collect and generate more data. Nevertheless, these issues greatly reduce**
**my confidence in the validity of the results, and I would recommend that the**
**authors address this issue explicitly in the text, as well as include analyses that show**
**that the below-standard dataset is not influencing the result. Maybe some sort of a**
**subsampling analysis, simulations, or other (newer) publicly available datasets used**
**as a point of comparison, to show the results using this small dataset are unbiased?**

We agree with the reviewer that we could increase the sequencing depth of these
samples. However, we do not think that undersampling questions the validity of the
results that we present here. Undersampling would only tend to destroy any underlying
phylogenetic signal, and not create false associations with phylogeny or diet where there
is none. The diversity that we have with these data is representative of the part of the
microbiome containing the most abundant bacterial taxa in each of these mammals. We
expect these most abundant bacteria to be involved in numerous functions related to host
diet or host metabolism/physiology. In addition, as suggested by the reviewer, we have
performed our BDTT analyses, our estimations of gain and loss of lineages and our co-
speciation analyses on rarefied OTU tables, which represent subsamplings of the initial
dataset. In all of these subsampling analyses, we reached strong and significant
conclusions, demonstrating that we have enough statistical power with these data to
discriminate alternative hypotheses.

In conclusion, although this dataset might not be the most exhaustive sampling of gut
bacteria in mammals, it is sufficient to capture strong, significant and coherent signals
regarding the dynamics of gut microbiome evolution that we observe and report for the
first time.

**2. Another problematic issue is that only a single individual is sampled from each**
**species. Thus, this analysis cannot account for within-species variation, which could**
**have a large effect, as we know from human studies. As above, I would encourage**
**the authors to address this and alleviate these concerns. Maybe using other available**
**datasets that contain multiple individuals from each species to show the selection of**
**a single individual from each species doesn't bias the result?**

We thank the reviewer for this comment. However, we do not think that the microbiome
compositional variability between individuals of the same species is a strong concern
regarding our main conclusions. The main reason is that in the case where intra host
variability would be higher than inter host variability, it would blur or break the signal

between host phylogeny or diet and microbiome composition. As we observe strong
associations between these variables at the inter host level, it means that the intra-host
compositional variance inherent to each host species has weak effects compared to inter-
host compositional variance. To support this point even further, we have controlled for
this effect with our data. The initial Muegge dataset included replicate samples for 7
hosts (Baboon, Big Horn, Human, Chimp, Hyrax, Lion and Okapi). We initially selected
only one individual for each of these species to focus on inter-host species comparisons.
We have substituted these 7 individuals with their conspecific and we have re-processed
the data with the exact same parameters. We computed the BDTT profiles characterizing
the correlation between the new microbiome compositional dissimilarities and host
phylogenetic or dietary distances (which remain unchanged). We now present these
results in Supp. Fig. 6 (below) and show that our initial conclusions hold true with this
different host sampling, confirming that intra-host compositional variability does not blur
signals of inter-host compositional differences.

Supp. Fig. 6: Control for the impact of intra-host variability on the scale disparity between the effects of host phylogeny and diet.

The BDTT analyses were run as in Fig. 1A. The plain blue and orange lines show the original correlation profiles with host phylogeny and diet, respectively (Fig. 1A). The

547 dashed lines show the correlation profiles with both factors that we obtained when using
the gut microbiome of alternative individuals for 7 host species. This control confirms
that the intra-host compositional variability is much weaker than the inter-host
compositional differences and that our main conclusions regarding associations between
microbiomes and host phylogeny and diet drawn in our manuscript are not biased by our
choice of individuals within each host species.

**3. I might have missed it, but I could not find a link to view or downloaded the data**
**used here. These should be made freely available to reviewers and readers who may**
**be interested in replicating the study or re-analyzing the data for new biological**
**findings. Ideally, there should be a link to download all the processed datasets used**
**in the analysis.**

This is a good point. The link to download the original data can be found in Muegge et al.
paper. We have added an additional link to download the multiple sequence alignment of
the processed 16S data that we used for all phylogenetic analyses (lines 442-449).
Furthermore, we have also deposited the OTU table of unique sequences, the calibrated
and non-calibrated bacterial phylogenetic trees, the matrix of host phylogenetic distances
and the matrix of host dietary distances.

REVIEWERS' COMMENTS:

Reviewer #3 (Remarks to the Author):

Thank you very much for seriously considering all of my comments and acting on most of them.

Reviewer #4 (Remarks to the Author):

The authors have adequately addressed my concerns.

REVIEWERS' COMMENTS:

Reviewer #3 (Remarks to the Author):

Thank you very much for seriously considering all of my comments and acting on most of them.

We thank the reviewer for helping us improving greatly the manuscript.

Reviewer #4 (Remarks to the Author):

The authors have adequately addressed my concerns.

We thank the reviewer for helping us improving greatly the manuscript.